# Legume Pangenome: Status and Scope for Crop Improvement

**DOI:** 10.3390/plants11223041

**Published:** 2022-11-10

**Authors:** Uday Chand Jha, Harsh Nayyar, Eric J. B. von Wettberg, Yogesh Dashrath Naik, Mahendar Thudi, Kadambot H. M. Siddique

**Affiliations:** 1Indian Institute of Pulses Research, Kanpur 208024, India; 2Department of Botany, Panjab University, Chandigarh 160014, India; 3Department and Plant and Soil Science, Gund Institute for the Environment, The University of Vermont, Burlington, VT 05405, USA; 4Department of Agricultural Biotechnology and Molecular Biology, Dr. Rajendra Prasad Central Agricultural University, Pusa 848125, India; 5Shandong Academy of Agricultural Sciences, Jinan 250100, China; 6Department of Agricultural Biotechnology and Molecular Biology, University of Southern Queensland, Toowoomba, QLD 4350, Australia; 7The UWA Institute of Agriculture, The University of Western Australia, Perth, WA 6001, Australia

**Keywords:** pangenome, legumes, structural variation, genome sequence, gene, crop domestication

## Abstract

In the last decade, legume genomics research has seen a paradigm shift due to advances in genome sequencing technologies, assembly algorithms, and computational genomics that enabled the construction of high-quality reference genome assemblies of major legume crops. These advances have certainly facilitated the identification of novel genetic variants underlying the traits of agronomic importance in many legume crops. Furthermore, these robust sequencing technologies have allowed us to study structural variations across the whole genome in multiple individuals and at the species level using ‘pangenome analysis.’ This review updates the progress of constructing pangenome assemblies for various legume crops and discusses the prospects for these pangenomes and how to harness the information to improve various traits of economic importance through molecular breeding to increase genetic gain in legumes and tackle the increasing global food crisis.

## 1. Introduction

Global climate change causing significant crop yield losses, burgeoning human population growth, and increasing evidence of malnutrition and hunger-related issues are serious challenges worldwide [1,2]. Novel scientific interventions are urgently needed to minimize these rising challenges and ensure global food security by developing climate-resilient food crops [3]. Advances in genomics resources, such as molecular markers, have greatly expanded the plant breeding tools for improving various plant traits to meet the global human food demand [3,4]. The paradigm shift and technological advances in DNA-sequencing technology, especially next-generation sequencing (NGS), have aided the construction of accurate reference genome assemblies for various legume crops, allowing the identification of an underlying target-trait candidate gene(s), structural variations, and genetic diversity and insights into domestication and evolutionary history [5,6,7]. However, a single reference genome assembly is insufficient to capture all the structural variations, nucleotide variations, and gene-content information for all individuals related to a particular crop species [8]. Therefore, species-representative genomes or ‘pangenomes’ are needed to capture all of the structural variations (SVs), including presence–absence variations (PAVs), copy number variations (CNVs), and repetitive elements or transposable elements (TEs), across the entire genome of all individuals in a plant species [8,9,10,11,12,13,14,15,16]. Pangenomes contain ‘core genes’ common to all the individuals of a species and ‘accessory genes’ or ‘variable genes’ present in some but not all individuals [9]. Accessory genes vary more than core genes, enabling plant adaption under diverse environments [17].

Tettelin et al. [9] first proposed the pangenome concept for *Streptococcus agalactiae* bacteria. Subsequently, Morgante et al. [18] reported this concept in plants. Li et al. [17] constructed the first pangenome of soybean to explore the role of various gene(s)/genomic regions contributing to agronomic traits and the domestication process and to track evolutionary history. The pangenomes of various plant species have been constructed, including Arabidopsis [19], rice [20], wheat [21], maize [22], Brachypodium [23,24], sorghum [25], brassica [26,27], sunflower [28], tomato [29], pepper [30], cotton [31], and apple [32], to unravel gene-content variation/genome dynamics, domestication, evolution processes, and the gene(s) lost during selection and breeding activity.

Legume crops play a crucial role in ensuring global food security. Legume reference genomes have been constructed for soybean [33], chickpea [34], common bean [35], pigeonpea [36], pea [37,38], lupin [39,40], peanut [41], cowpea [42], and mungbean [43]. However, pangenomes are needed to obtain insights into the genome dynamics, gene-content variation, genetic basis of agronomic-trait variation, and evolutionary relationship in various legumes. This review discusses the current status of various pangenomes available in various legume species and how they can be exploited in genomics-assisted trait improvement to increase their genetic gain.

## 2. Pangenome Capturing Structural Variations Not Present in Single Reference Genomes

Plant genomes vary between individual genotypes within the same species due to variations in gene content, including CNVs and PAVs, inversions, and the presence of transposable elements (as seen in [44,45,46]). Several mechanisms cause structural variations in genomes. The replication ability of transposable elements (TEs) allows these TEs to change position across the whole genome and insert to any coding region, functionally disrupting these genes [47,48]. Recently, a pangenome-sequence analysis of peas estimated 85.5% of deletions and 77.4% of duplications in long terminal repeat (LTR) Copia and LTR Gypsy sequences [38]. Thus, these transposable elements cause insertions and deletions in genomes, rendering changes in plant phenotypes [46,49]. Likewise, unequal recombination or crossover can introduce SVs in genomes [50]. The other important mechanisms for introducing SVs are segmental genome duplication and whole genome duplication found in the cases of soybeans [33] and peanuts [41]. Inversion and transversion and chromosomal translocations are other mechanisms accounting for genome SVs [6,7,15,51]. Thus, a single reference genome assembly does not capture the entire gene repertoire of a species. In contrast, pangenomes harbor core and dispensable genes encompassing all structural variations in non-coding and coding sequences of all individuals of a particular species (Figure 1). Pangenome analysis can be performed using three approaches: de novo assembly, reference-based assembly and iterative mapping, and graph-based assembly [13,15,46,52].

## 3. Soybean (*Glycine max* L.)

In soybean, Li et al. [17] performed pangenome analysis by constructing a de novo genome assembly of seven accessions of *Glycine soja* collected from different geographic regions to shed light on the genetic basis of untapped genetic diversity and evolution-related novel insights. Comparing the genome sequences obtained from this pangenome analysis and a reference genome sequence of *G. max* W82 revealed a plethora of loss or gain of CNVs in multiple genes, ranging from biotic-stress tolerance (including R genes) to various transcription factors contributing to local-environmental acclimation against invading pathogens [17]. In addition, comparing the *G. max* and *G. soja* genomes provided novel insights into the *FLOWERING LOCUS T* gene controlling flowering time, *PHYA*-regulating light receptor, and *LEAFY* genes via various ‘indels’ and nonsynonymous SNPs in the given loci in the *G. soja* genome [17]. In *Arabidopsis*, 1332 genes related to seed protein and oil contents were annotated in the *G. max* W82 genome based on homology to genes related to lipid-acyl pathways [53]. A comparative genomics analysis revealed that the *G. max* and *G. soja* genomes differ for several indels and large-effect single-nucleotide variants (SNPs) and CNVs for these genes attributing to oil-seed content [17]. A de novo assembly of 26 soybean accessions (including 14 cultivated, 3 wild, and 9 landraces) from 2898 deeply sequenced soybean genomes and pangenome analysis provided novel insights into the thousands of structural variations related to various candidate genes of agronomic interest (including iron-deficiency chlorosis, seed luster, and flowering) and involved in domestication and evolution [54] (Figure 2). The assembled sequences contained 20,623 gene families designated as core genes, with 8163 gene families in 25–26 accessions designated soft-core genes, and 28,679 families in 2–24 accessions designated dispensable genes [54] (Table 1). 

A comparative analysis of this pangenome with the previously assembled reference genomes for Zhonghuang [63,64], W05 [65], and Wm82 [33] indicated the presence of myriad PAVs and CNVs in various accessions of current soybean accessions, demanding a thorough pangenome analysis of diverse and geographically distinct soybean accessions [54]. Torkamaneh et al. [16] developed the soybean pangenome known as ‘Pansoy’ by constructing a de novo assembly of 204 geographically diverse soybean accessions following the ‘map to pan’ approach using the EUPAN pipeline [66]. This pangenome contained 108 Mbp novel genome sequences—with 1659 novel genes not present in the reference Wm82 genome—revealing 49,431 hard-core genes, 1401 soft-core genes, 3402 shell genes, and 297 cloud genes [16]. Thus, the Pansoy resource provided novel insights into intraspecific genetic variation in *G. max* and offered a platform for genomic diversity and evolutionary and domestication studies for soybeans.

Subsequently, a pangenome measuring 1213 Mbp was constructed from >1000 accessions collected from the USDA Soybean Germplasm collection using the chromosome level ‘Lee’ soybean genome assembly as the reference sequence [55]. The results identified 3765 genes missing in the ‘Lee’ genome sequence assembly, indicating a loss of genetic diversity during domestication and breeding activity [55]. This pangenome also contained an additional 198.34 Mbp sequence assembly compared to the ‘Lee’ reference genome. The PAV analysis revealed 86.8% core genes and 13.2% dispensable genes [55]. The authors also uncovered 110 genomic regions containing 1266 protein-coding genes in a ‘domestication-selective sweep’ and 86 genomic regions containing 1434 genes in a ‘breeding-selective sweep’ [55]. Based on this pangenome analysis, the authors concluded that wild soybean had higher nucleotide diversity than landraces, old cultivars, and modern cultivars due to the loss of diversity during domestication and breeding efforts.

## 4. Chickpea (*Cicer arietinum* L.) Pangenome

Chickpea is a major pulse crop, ranking third among pulses for global production and serving as a plant source of protein and many essential micronutrients [59]. In the last decade, unprecedented advancements in bioinformatics and NGS technologies allowed for the assembly of a chickpea genome sequence [34,67], which has served as a tool for chickpea improvement. To further build chickpea-genomic resources and to study speciation, phylogenetic analysis, genomic diversity of cultivated species and its wild progenitor, and migration of cultivated chickpea species, Varshney et al. [59] constructed the first chickpea pangenome (Figure 2) (measuring 592.58 Mb with 29,870 genes) by sequencing 3171 cultivated and 195 wild chickpea accessions using iterative mapping and an assembly approach. From this, 124,833 SNPs were selected to form linkage-disequilibrium blocks; these SNPs and the phenotyping data for 100 seed weight (100 SW) and yield per plant (YPP) from 2980 genotypes were used for genomic prediction to improve these traits [59]. Likewise, 3.94 million SNPs and the phenotyping data for 16 phenological and yield-related traits in 2980 genotypes revealed associations between 205 SNPs and 11 traits, with 152 SNPs associated with 79 unique genes governing seed size and development [59].

## 5. Cowpea (*Vigna ungiculata* L.) Pangenome

Cowpea is an important warm season, climate-resilient, multipurpose legume crop in sub-Saharan Africa [68]. Significant progress has been achieved in developing genomics resources for cowpeas. Lonardi et al. [42] developed the genome assembly of cowpea genotype IT97K-499-35. Liang et al. [57] constructed the first cowpea pangenome by producing de novo assemblies of six cowpea accessions with a mean size of 449.91 Mb, identifying 21,330 core and 23,531 non-core genes, with the large number of frameshift variations in the non-core genes attributing large diversity in domesticated cowpeas (Table 1). Comparative and genomic synteny analyses of the IT97K-499-35 reference genome and the genome assemblies of the six accessions indicated the presence of numerous structural rearrangements, including PAVs, inversions, indels, and CNVs [57]. The authors confirmed that the existing PAVs are responsible for the black seed-coat color in cowpeas (Table 2).

## 6. Pigeonpea (*Cajanus cajan* L.) Pangenome

Pigeonpea is an important grain legume, mostly grown in India and Africa, with an essential role in ensuring protein nutrition security across the semi-arid tropics [36]. The pigeonpea reference genome sequence [36] has been used to improve various traits in pigeonpea; however, the pangenome assembly of pigeonpea provides further opportunities to harness untapped genetic variation not present in the reference genome sequence. Zhao et al. [58] built the pigeonpea pangenome (measuring 622 Mbp) using iterative mapping and an assembly approach [13], sequencing 89 pigeonpea accessions collected across the globe and the reference pigeonpea with >9.5× coverage. The pangenome contained 55,512 more genes than the previously assembled reference genome (53,612), with 48,067 core genes and 7445 accessory genes [58]. A functional analysis of the ‘variable genes’ indicated their possible role in proteolysis, pollen tube reception, and the signal transduction process [58]. Of 909 identified R genes, 836 were core genes and 73 were variable genes. To confirm the role of PAVs attributing phenotypic diversity, the authors detected several additional SNPs for various phenotypic traits (seed weight, days to 50% flowering, and plant height) compared to Varshney et al. [70], who found a significant association of 1 SNP for pods/plant, 1 SNP for primary branches/plant, and 3 SNP for secondary branches/plant from the study of whole genome resequencing of 292 pigeonpea accessions.

## 7. Mungbean (*Vigna radiata* L.) Pangenome

Mungbean is an important grain legume crop with high nutritional benefits, mostly grown in south and southeast Asian countries [71]. Draft genome assemblies are available for mungbean genotypes VC1973A [43] and JL7 [56]. Liu et al. [56] assembled the first mungbean pangenome (measuring 762.92 Mb), with 43,462 annotated genes, by deep-sequencing JL7 and 217 mungbean accessions, including cultivars and landraces. The predicted genes comprised 33,258 hard-core genes, 2872 soft-core genes, 7154 shell genes, and 178 cloud genes [56]. Of the nine PAVs controlling flowering regulation, three genes (*jg13350, jg13746*, and *Pang80812*) were present in Chinese landraces and breeding lines [56], while six genes (*jg1521*, *jg5273*, *jg5274*, *jg5281*, *jg5284*, and *Pang68295* were absent but present in non-Chinese lines [56].

## 8. White Lupin (*Lupinus alba* L.) Pangenome

White lupin (*Lupinus albus* L.) is a protein-rich grain legume crop domesticated in the Mediterranean region [72,73]. However, the presence of quinolizidine alkaloids in lupin seed causes a bitter taste and toxicity; thus, lowering seed alkaloids is a prime objective of lupin-breeding programs. Hufnagel et al. [40] and Xu et al. [74] constructed the white-lupin genome assembly. Hufnagel et al. [61] established the first white-lupin pangenome by sequencing a diverse set of 39 white-lupin accessions using a ‘map-to-pan’ approach [66]. The pangenome comprised 32,068 (78.5%) core genes, 6046 soft-core genes, and 8776 shell genes [61] (Table 1). Several selection sweeps related to breeding for low-alkaloid accessions were identified on chromosomes 18, 15, 1, and 3, overlapping the low-alkaloid-content QTLs previously reported by Ksiazkiewicz et al. [75] and Lin et al. [76].

## 9. Barrel Medic (*Medicago truncatula*) Pangenome

Barrel medic **(***Medicago truncatula*) is an important model legume crop used for investigating symbiotic relationships with rhizobia and mycorrhizae and root development [77,78]. Zhou et al. [60] developed the first *M. truncatula* pangenome (measuring 388–428 Mbp, with an additional 63 Mbp novel sequence) by de novo assembly of 15 *M. truncatula* accessions, adding 16% genome space to the reference genome sequence [79] of this model legume (Table 1). This pangenome sheds light on causative structural variants, including transposable elements in different gene families, such as the nucleotide-binding site leucine-rich repeat rapid evolvement.

## 10. Lentil (*Lens culinaris*) Pangenome

Lentil is an important source of dietary protein, carbohydrates, and iron for humans [80]. Due to the large size of the genome (4 Gb) and the presence of a high number repetitive elements, the completion of a lentil genome assembly has taken some time to complete. However, an incomplete-draft genome sequence of *Lens culinaris* cv. CDC Redberry v2.0 [81,82] has facilitated new directions in lentil genomics research. Thus, in the absence of a complete reference genome sequence, pan-transcriptomes [83] could provide a platform to establish the lentil pangenome. Guerra-Garcia et al. [84] have used an exome capture array built on the draft lentil genome and transcriptomic data [85] to examine copy number variation across cultivated lentil germplasm. This work has uncovered a previously hidden amount of variation across lentils [84]. Furthermore, Gutierrez-Gonzalez et al. [62] assembled and annotated the transcriptomes of eight lentil accessions, including cultivated and wild species (*L*. *orientalis*, *L*. *tomentosus*, *L*. *ervoides*, *L*. *lamottei*, *L*. *nigricans*, and two *L*. *odemensis*), identifying 15,910 core genes and 24,226 accessory genes. A comparative analysis of these assembled transcriptomes with the incomplete CDC Redberry reference genome assembly indicated that transcripts of *L. culinaris* had the greatest similarity with CDC Redberry transcripts, while transcripts of *L*. *lamottei* and *L*. *nigricans* had the least [62]. Further long-read-based sequencing efforts could assist in decoding the complete genome and pangenome assembly of lentils.

## 11. Pea (*Pisum sativum* L.) Pangenome

Pea is the fourth-most globally important grain legume, enriched with protein, starch, and minerals [38,86], and is considered a ‘founder crop’ member; it is also one of the oldest domesticated crops [87]. Despite having an assembled pea reference genome [37], genetic diversity in cultivated and wild pea and the domestication history of pea remain elusive. Yang et al. [38] constructed a second pea de novo genome assembly of ZW6 and a pangenome assembly of 116 accessions containing cultivated and wild pea genotypes, identifying 112,117 pangenes, including 15,470 core genes, 6170 soft-core genes, 41,028 shell genes, and 50,108 cloud genes. The study confirmed that *P. abyssinicum* is separate from *P. fulvum* and *P. sativum.* The pangenes from *P. abyssinicum* were unique in stimulus and chemical response, while those of *P. fulvum* were involved in growth, tropism, and development; thus, the novel genes harboring in these species could be used to improve disease resistance and yield in modern pea cultivars [38].

## 12. Pangenome-Derived PAVs for GWAS Analysis to Explore Novel Genes

Pangenome analysis allows the discovery of a plethora of novel SVs, including gene PAVs that may be missing in the reference genome assembly [16,27,56]. Thus, these pangenome-derived PAVs can be used in a GWAS to uncover a novel agronomic trait gene allele(s). Liu et al. [56] uncovered five novel genes missing in the mungbean reference genome assembly that contributed to bruchid resistance (*Pang34265*, *Pang44622*, *Pang57772*, *Pang58608*, and *Pang64254*). Zhao et al. [58] identified 225 significant SNPs associated with various phenological and yield-related traits (e.g., days to 50% flowering, days to maturity, pods/plant, and yield/plant) from SNPs and PAVs derived from a pigeonpea pangenome analysis, which overlapped those previously identified by Varshney et al. [70]. However, several new SNPs were associated with these studied traits [58]. Likewise, Varshney et al. [59] identified 205 significantly associated SNPs for agronomic traits in a GWAS study on 16 traits in a panel of 2980 chickpea accessions using 3.94 million SNPs, of which 152 SNPs were in 79 unique genes related to seed size and seed development [59]. GWAS analysis using SVs derived from 2898 soybean accessions elucidated the genetic basis of soybean seed luster, uncovering the presence of 10 kb PAVs in the hydrophobic protein-encoding gene causing seed luster [54].

## 13. Pangenome and Scope of Crop Domestication for New Species

The availability of reference genomes and pangenome assemblies in various legume crops facilitates identifying causal variants, such as SVs, related to domestication traits (Figure 3). Genome-editing tools allow us to edit these traits from wild species to make them agronomically desirable [45]. For instance, pangenome analysis indicated that *G. soja,* a wild-crop relative of soybean, contained more R-gene domain architectures (25) than the cultivated counterpart *G. max* (14), providing increased advantages against various biotic stress [17]. Similarly, *GmFT2c,* a candidate gene regulating flowering time, was present as a complete gene structure in *G. soja*; however, GmaxW82 had *GmFT2c* gene with missing up to 50% N-terminal peptide [17]. Pangenome analysis revealed that the change in seed coat color from black in wild soybean cultivars to yellow in cultivated cultivars related to SV inversion in the chalcone synthase gene (*CSH*) and *CSH* gene duplication during soybean domestication [54]. Likewise, the authors confirmed that wild soybean contains more SVs than landraces and cultivated soybean from 26 de novo genome assemblies and data for 2872 soybean lines. Bayer et al. [55] also reported gene losses during domestication in a pangenome analysis of >1000 soybean lines, with wild soybean containing more genes (48,785) than landraces (48,371) and modern cultivars (48,165). In another pangenome analysis, Varshney et al. [59] indicated that present-day chickpea cultivars contain several deleterious alleles due to domestication, affecting genetic fitness. Intensive breeding activities and domestication have reduced genetic diversity [55,59]. North American soybean breeding lines possess an estimated 85% of the genes derived from only 19 landraces of soybean [88]. In another study, 79% of rare alleles harbored by diverse landraces were lost during breeding in soybean [89]. A pangenome analysis indicated that wild-pea genomes (e.g., *P. fulvum* and *P. abyssinicum*) contained higher SVs than the cultivated species (*P. sativum*), confirming that wild genomes are rich in novel genes of agronomic traits [38]. Thus, genetic variants differ between cultivated and wild types, and the genome editing could target deleterious alleles for domestication [45].

Particularly useful targets of breeding for de novo domestication or for enhanced utilization of wild relatives of existing crops come from our growing understanding of the genetic basis of domestication in some of the most extensively studied legumes. The domestication syndrome has long been a useful concept for this, with a few essential traits repeatedly selected by farmers to convert wild plants into crops that could be effectively planted, harvested, and consumed (e.g., [90,91,92,93,94]). Cultivated legumes share similarities in indehiscent pods [95,96], loss of seed dormancy [97], increased seed size (e.g., [98,99]), and reduced defensive chemistry (e.g., [100,101]). These have long been suspected to have a common genetic basis (e.g., [102]). Furthermore, post-domestication traits involved in crop spread from centers of origin and diversification into distinct agro-ecologies may also have shared genetic bases (e.g., [103]). Phylogenetically informed super-pangenomes are the ideal tool to examine questions of repeated selection on particular loci, pathways, or gene families across legumes (e.g., [15]). In legume clades with multiple domestications, such as the genera *Vigna, Phaseolus, Lupinus*, and the closely related cool-season legumes in *Pisum, Lens,* and *Vicia,* genus or even tribe legume pangenomes would be immense tools for understanding shared and contrasting patterns of selection. These tools may also assist in harnessing germplasm for crop improvement. In forage legumes, such as *Medicago sativa* and *Trifolium*, these tools could have a similar impact [78]. Additionally, in legumes that are potential targets of neodomestication, such as legumes making industrially useful compounds like *Derris, Tephrosia,* and *Indigofera*, these tools could be equally powerful.

## 14. Prospects and Conclusions

The increasing human population, decreasing land for agriculture, and drastic changes in global climate change pose serious challenges for crop production to ensure food security [3,4], requiring novel breeding and genomics approaches. Pangenome resources of major legume species could be instrumental in shedding light on crop domestication and evolution and interrogating the genetic basis of agronomic traits to improve genetic grain (Figure 3). Further, these pangenome resources could be used for developing novel single-nucleotide polymorphism, performing GWAS, mapping QTL, mining superior haplotypes of various traits, and genomic prediction in crop breeding programs [15,26,52,104]. Combining pangenome information with CRISPR-Cas9 based genome-editing technology could open up new avenues for de novo domestication of wild species [45]. Leveraging pangenome, transcriptome, epigenome, metabolome, and phenome data with machine-learning approaches could greatly benefit genomics-assisted breeding [104]. Further, these resources could help improve underutilized legumes with no developed reference genome or pangenome assembly. Further advances in long-read sequencing and improved genome-assembly construction and genome annotation will help develop high-quality pangenomes with few gaps and assist in predicting PAVs in targeted trait genes with more precision [105,106]. With decreasing sequencing costs and advances in computational biology, including various software and analysis pipelines, pangenome studies could encompass all species in a genus to construct a ‘super pangenome’ [8,15], providing a comprehensive repertoire of genomic information at the genus and even family level.

## Figures and Tables

**Figure 1 plants-11-03041-f001:**
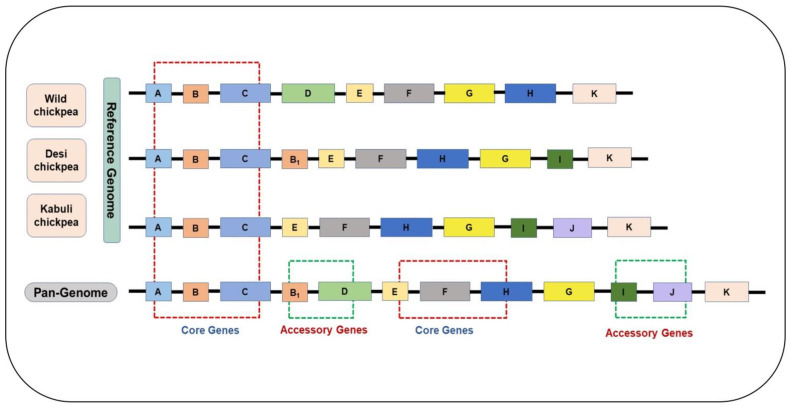
Phylogenetically diverse accession selected for construction of pangenome that contains core genes and accessory genes. Pangenome contains all the possible structural variation that may occur during evolution, such as B_1_: copy number variation, D: deletion variation, I and J: insertional variation, and G: transposable element.

**Figure 2 plants-11-03041-f002:**
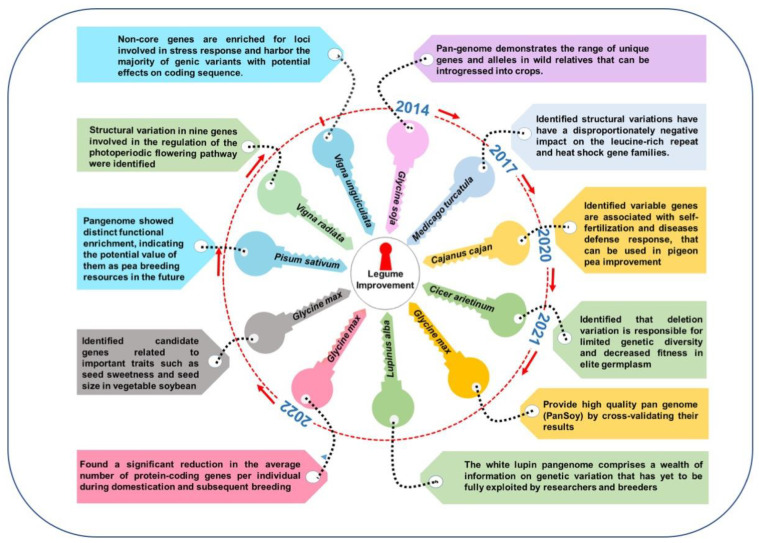
Legume pangenomes and key learnings for crop improvement.

**Figure 3 plants-11-03041-f003:**
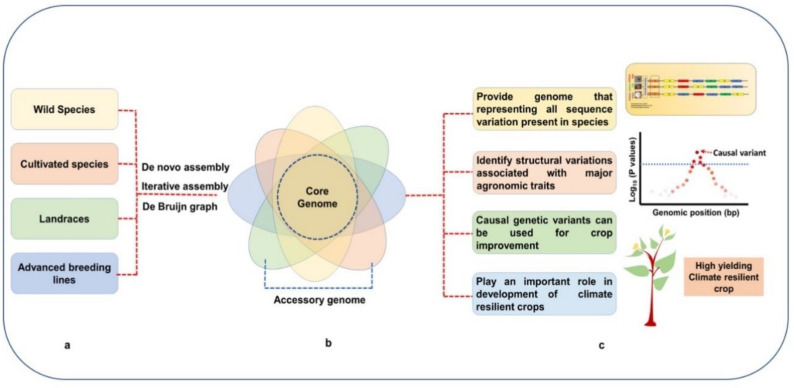
Summarizing pangenomes and their application in crop improvement: (**a**) different recourses and various assembly approaches for pangenome construction, (**b**) representative pangenome with shared (core genome) and non-shared genome (accessory genome) between the accessions, and (**c**) mention of the application of pangenome in crop improvement.

**Table 1 plants-11-03041-t001:** List of various pangenomes constructed in various legume crops.

Crop	Size of Pangenome	Number of Core and Dispensable Genes	Accessions Used	Reference
Wild soybean (*Glycine soja)*	889.33–1118.34 Mb	48.6% of the gene families	7	[17]
		were core genomic units and 51.4% of gene families (30,364) were non-core		
Soybean (*Glycine max)*	108 Mb	49 431 (90.6%) hard-core genes, 7.2% variable genes	204	[16]
Soybean (*Glycine max)*	992.3 Mb to 1059.8 Mb	20,623 core genes and 28,679 dispensable genes	26	[54]
Soybean (*Glycine max)*	1213 Mbp	86.8% of genes were core and 13.2% were dispensable	1110	[55]
Mungbean (*Vigna radiata)*	762.92 Mb	83.1% were core genes and 16.9% were variable	217	[56]
Cowpea (*Vigna ungiculata)*	449.91 Mb	21,330 core and 23,531 non-core genes	6	[57]
Pigeonpea (*Cajanus cajan)*	622 Mbp	48 067 (86.6%) core genes and 7445 (13.4%) non-core	89	[58]
Chickpea (*Cicer arietinum)*	592.58 Mbp	-	3366	[59]
Barrel medic *(Medicago turcatula)*	388 Mbp to 428 Mbp	250 Mbp core sequence and 180 Mbp dispensable	15	[60]
		sequence		
White lupin (*Lupinus alba)*	462.7 Mbp	32,068 core genes, 14,822 non-core genes	39	[61]
Lentil (*Lens culinaris)*	-	15,910 core genes, 24,226 accessory genes	8	[62]
Pea (*Pisum sativum)*		15,470 core genes, 6170 softcore genes, 41,028 shell genes, and 50,108 cloud genes	116	[38]

**Table 2 plants-11-03041-t002:** Applications of pangenome in various legume crops.

Crop	Salient Features	Reference
Wild soybean (*Glycine soja)*	2.3–3.9 Mbp of *G. soja* specific PAV related to defence response, cell growth, and photosynthesis. Pangenome analysis	[17]
	informed variation for protein, oil, flowering time, and organ-size traits in the wild and cultivated soybean species	
Soybean (*Glycine max*)	PanSoy sheds novel insights into the intraspecific variation in *G. max*	[16]
Soybean (*Glycine max*)	The function of core genes were related to growth, immune system, reproductive, cellular, and cellular-component organization or biogenesis.	[54]
	The function of non-core genes were related to abiotic and biotic response genes, such as different NBS gene families. Structural variation related to	
	domestication traits and agronomic trait, viz., iron-deficiency chlorosis	
Soybean (*Glycine max*)	Domestication selection sweeps on chromosome Gm20, breeding-related selective sweep region on Gm20,	[55]
	110 genomic regions with signatures of domestication-selective sweeps harboring 1266 protein-coding genes,	
	86 genomic regions with signatures of breeding-selective sweeps harboring 1434 protein-coding genes	
Mungbean (*Vigna radiata*)	A total of nine presence/absence variations controlling flowering regulation. Bruchid-resistant genes *Pang34265*, *Pang44622*, *Pang57772*, *Pang58608*, and *Pang64254*	[56]
Cowpea (*Vigna ungiculata)*	PAVs contributing to black seed-coat color in cowpeas	[57]
Pigeonpea (*Cajanus cajan)*	PAVs attributing phenotypic diversity, including various phenotypic traits, viz., seed weight, days to 50% flowering, and plant height	[58]
Chickpea (*Cicer arietinum)*	643 gene-gain and 247 gene-loss CNVs in *C. reticulatum* accessions,	[59]
	insertions (139,483), deletions (47,882), inversions (61,171),	
	intra-chromosomal translocations (417) and inter-chromosomal translocations (2410) in cultivated and 287,854 insertions,	
	67,351 deletions, 58,070 inversions, 446 intra-chromosomal translocations and 2066 inter-chromosomal translocations among *C. reticulatum* accessions	
Barrel medic (*Medicago truncatula*)	500,000–1,500,000 short indels (<50 bp), 27,000–110,000 large indels, 49,000–169,000 copy number variants (CNVs),	[60]
	and 2700–12,700 translocations; NBS-LRRs showed high SNP diversity;	
*Medicago sativa*	circRNA, lncRNA responsive to salinity and drought	[69]
White lupin (*Lupinus alba)*	Alkaloid-related genes/QTLs/genomic region, 1195 PAVs	[61]
Lentil (*Lens culinaris)*	Comparative analysis and evolutionary analysis, transcriptome of *L. culinaris* contained genes 58,375	[62]
	*L. nigricans* transcriptome contained the minimum number of 46,742 genes	
Pea (*Pisum sativum)*	Two seed-dormancy genes *Psat02G0081200* and *Psat02G0507900* were elucidated; pan genes of *P. abyssinicum* were unique in response to chemicals and stimuli;	[38]
	pan genes of *P. fulvum* showed their role in development, cytoskeleton, and tropism	

## Data Availability

The data presented in this study are available in the article.

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
