# Peer review of "Legume Pangenome: Status and Scope for Crop Improvement"

_plants, 2022, doi:10.3390/plants11223041_

Round 1
Reviewer 1 Report
This review paper on legume pangenomes is well written and also reads well. The topic is covered extensively, with appropriate figures and tables to support the text. There is a logical flow in the paper. There are just quite a number of typos, language errors and small editorial issues that I have indicated on the attached pdf that need to be corrected. Also just check, you generally wrote pangenome, but in some places you wrote pan-genome. Just standardize.

Author Response
Reviewer1
This review paper on legume pangenomes is well written and also reads well. The topic is covered extensively, with appropriate figures and tables to support the text. There is a logical flow in the paper. There are just quite a number of typos, language errors and small editorial issues that I have indicated on the attached pdf that need to be corrected. Also just check, you generally wrote pangenome, but in some places you wrote pan-genome. Just standardize.
Response
We thank the reviewer for positive and constructive suggestions for improving the manuscript. We have kept uniformity for writing pangenome highlighted in green throughout the text. The other correction raised are also highlighted in green colour in the text.

Reviewer 2 Report
In this review, the authors reviewed the origin and development of idea of a pan-genome. Then they summarized the status of pan-genome constructions in nine different Legume species, including soybean, chickpea, cowpea, pigeonpea, mung bean, white lupin, barrel medick, lentil, and pea. Lastly, they discussed how pan-genome studies can bring insights into crop breeding. Overall, the manuscript is well written. My only suggestion is to include both scientific names and common names for each of the species mentioned in the manuscript. For example, on tables 1 and 2, please add the common name of each crop in parentheses following its scientific name. For the subtitles, several plants, such as chickpea and cowpea were referred by their common names, while Glycine max L and Medicago truncatula were referred by their scientific names. It would be helpful to consistently use either the common names or the scientific names when referrer to all species.
Author Response
Reviewer2
In this review, the authors reviewed the origin and development of idea of a pan-genome. Then they summarized the status of pan-genome constructions in nine different Legume species, including soybean, chickpea, cowpea, pigeonpea, mung bean, white lupin, barrel medick, lentil, and pea. Lastly, they discussed how pan-genome studies can bring insights into crop breeding. Overall, the manuscript is well written. My only suggestion is to include both scientific names and common names for each of the species mentioned in the manuscript. For example, on tables 1 and 2, please add the common name of each crop in parentheses following its scientific name. For the subtitles, several plants, such as chickpea and cowpea were referred by their common names, while Glycine max L and Medicago truncatula were referred by their scientific names. It would be helpful to consistently use either the common names or the scientific names when referred to all species
Response
Response
We have used common name and scientific name of the all legumes in the manuscript and in tables. In subtitles we have used common name and scientific name in bracket please see the yellow highlights in both text and in tables.
